# Abscisic Acid May Play a Critical Role in the Moderating Effect of *Epichloë* Endophyte on *Achnatherum inebrians* under Drought Stress

**DOI:** 10.3390/jof8111140

**Published:** 2022-10-28

**Authors:** Xuelian Cui, Wen He, Michael John. Christensen, Jinfeng Yue, Fanbin Zeng, Xingxu Zhang, Zhibiao Nan, Chao Xia

**Affiliations:** 1The State Key Laboratory of Herbage Seeds and Grassland Agro-Ecosystems, Key Laboratory of Grassland Livestock Industry Innovation, Ministry of Agriculture and Rural Affairs, Engineering Research Center of Grassland Industry, Ministry of Education, College of Pastoral Agriculture Science and Technology, Lanzhou University, Lanzhou 730000, China; 2AgResearch Limited, Grasslands Research Centre, Palmerston North 4442, New Zealand

**Keywords:** *Epichloë* endophyte, abscisic acid, drought stress, photosynthesis

## Abstract

Water scarcity is a major constraint that adversely affects plant development and growth. Abscisic acid (ABA) is a plant stress hormone that is rapidly synthesized and can induce stomatal closure to conserve water, thereby alleviating the drought stress of plants. The *Epichloë* endophyte enhances the drought tolerance of *Achnatherum inebrians* (drunken horse grass, DHG). To better understand how the *Epichloë* endophyte enhances drought tolerance, DHG plants without (EF) and with (EI), an *Epichloë* endophyte, were grown under 20% and 60% soil water conditions (SWC), and the leaves of the three treatments of EF and EI plants were sprayed with ABA solution (1 mg/L); fluridone (FLU), the ABA biosynthesis inhibitor solution (1 mg/L); and distilled water, respectively. Four-weeks later, the results indicated that the exogenous ABA application promoted plant growth, stomatal conductance, and photosynthetic rate, while the opposite effect occurred with plants sprayed with FLU. The differences between EI and EF plants in tiller number, height, chlorophyll content, stomata conductance, and photosynthetic rate were highest when sprayed with ABA. Thus, it is concluded that ABA might be involved in the moderating effect of *Epichloë* endophytes on DHG plants exposed to drought by maintaining growth and improving photosynthetic efficiency.

## 1. Introduction

To enhance growth and resistance under adverse environments, the associations between nearly all plants and a range of microorganisms, such as nitrogen-fixing bacteria, fungal endophytes, and mycorrhizal fungi, are considered as beneficial symbiotic relationships [1,2,3]. *Epichloë* endophytes exist in many cool-season grasses of the subfamily Pooideae; they live asymptomatically within the intercellular spaces of all tissues, except for the roots [4]. The host plants are symptomless and the transmission of many species in nature is entirely vertical through the seeds of the hosts [5]. The relationships between *Epichloë* endophytes and grasses are generally considered mutualistic [6]. Most studies focusing on *Epichloë* endophytes have involved grasses of the genera *Festuca* and *Lolium* [7]. Host plants provide photosynthetic products and a stable habitat for these *Epichloë* endophytes, and the endophyte presence can protect their hosts against abiotic [8,9,10] and biotic stresses [11,12] via the production and induction of fungal alkaloids and other secondary metabolites.

*Achnatherum inebrians*, commonly known as drunken horse grass (DHG), is a perennial bunchgrass of the Poaceae family. It is widely distributed in the arid and semi-arid grasslands of northwest China, where there is a low rainfall-drought climate [13]. Nearly 100% of wild DHG populations were infected with an *Epichloë* spp. [14]. Ergot alkaloids, including ergine and ergonovine, are produced in DHG plants infected with *Epichloë* endophytes [15], which can cause toxicity in grazing livestock and deter ingestion [16]. The presence of *Epichloë* endophytes benefits DHG plants by conferring increased growth and competitive ability, including protection from livestock grazing [17], insect pests [18,19], and pathogenic fungi [20,21,22]. Endophytes are also reported to improve the abiotic stress tolerance of host plants, including an increased tolerance of low temperature [23], heavy metals [24], salt stress [25,26], and, in particular, drought [27].

Photosynthesis is one of the critical primary physiological processes of plants [28] and is severely affected by drought stress [29]. Drought stress limits plant growth by reducing photosynthetic pigments, inhibiting the photosynthesis rate, and restricting the fixation of carbon dioxide [30]. There are thus huge beneficial impacts from improvements in the efficiency of photosynthesis of plants under drought conditions [31]. Phytohormones are the main signals under stress conditions, and almost all plant processes are directly or indirectly influenced [32]. Of special importance is ABA, which plays an important role in the plant response to a water deficit and has been considered to be one of the main internal plant signals that triggers the numerous acclimations that plants undergo when exposed to drought [33]. It has been shown to be involved in enhancing drought tolerance, both from the use of applied exogenous ABA to intact plants [34], and from the measurement of the endogenous ABA content [35]. Under drought conditions ABA rapidly accumulates, accelerating leaf senescence, inducing the biosynthesis of protective substances [36]. Substantial evidence provides support that ABA plays a pivotal role in the protection of plants during drought conditions by inducing stomatal closure and reducing shoot expansion [37,38,39]. In addition, ABA stimulates the elongation of the main root in response to drought [40], improving the plant water status and/or root water uptake [41]. Chlorophyll can be protected by ABA from degradation under drought stress [42]. The relative amount of chlorophyll is closely related to the photosynthetic capacity of plants [43]. Thus, ABA also plays a central role in the regulation of photosynthesis.

It has been proven that *Epichloë* endophytes increase the content of phytohormones, such as indoleacetic acid, in tall fescue (*Festuca arundinacea*) [44]. Further, it has been speculated that *Epichloë* endophyte infection can enhance the production of ABA under drought conditions, as the expression of genes encoding key enzymes involved in biosynthesis and signaling by ABA were upregulated in EI plants [45,46]. However, the impact of ABA in the enhancement of the drought tolerance of plants that host an *Epichloë* endophyte is poorly understood. In the present study, a controlled environment trial was employed to test whether ABA could be one of the substances involved in the moderating effect of *Epichloë* endophytes on DHG under limited water conditions through enhancing photosynthesis and growth.

## 2. Materials and Methods

### 2.1. Plant Material

As nearly 100% of DHG populations are infected with *Epichloë* spp. in nature, to obtain EF seeds, we started with the seeds from a single EI plant. These seeds were divided into two parts, with one portion of seeds treated with distilled water, and the other part treated with thiophanate-methyl fungicide to ensure that only non-viable hyphae were present in the seeds [47]. In the spring of 2014, these two seed populations were planted at spacings of 35 cm in the experimental field at the Yuzhong Campus of Lanzhou University (104°39′ E, 35°89′ N, altitude 1653 m). The endophyte-infection status of all plants that were established from the two seed populations was checked by a microscopic examination of leaf sheath pieces stained with aniline blue [48]. In the autumn of 2018, seeds were collected from the endophyte-free (EF) and endophyte-infected (EI) DHG plants, respectively, and stored at a constant 4 °C for the present study. In 2019, 10 seeds were selected randomly from both EI and EF seed sample to confirm the infection status before planting.

### 2.2. Experimental Design

In the present study, one polyethylene bag was placed into each of 120 plastic pots (height: 210 mm; diameter: 200 mm) and then 200 g of gravel was added to each bag. A tube (height: 220 mm; diameter: 10.5 mm) was inserted with one end exposed to the air while the other end was buried in the gravel at the bottom of the bag to ensure air exchange [27]. Two hundred grams of a 50:50 mixed medium of air-dried peat moss and vermiculite was added to every bag. The water-holding capacity of 200 g of mixed medium was measured and it could hold an average of 660 g of available water. Sixty of the 120 pots were used for the EF and EI plants, respectively. Three healthy-looking, well-filled seeds from the appropriate matching seed sample were sown into each of the medium-containing plastic bags. Each pot was later thinned to one seedling. All pots were assigned at random to a position within a constant temperature greenhouse (moisture: 76 ± 2%, temperature: 24 ± 2 °C with a photoperiod of 16/8 h (light/dark)) and were watered as needed.

After sixty days, the watering of the pots was ceased so that the water-holding capacity (WHC) of half of the EI and EF pots could be reduced to 20% relative soil water content (SWC)(drought), and the SWC of the other pots was reduced to 60% SWC (abundant moisture). Abscisic acid and fluridone (ABA biosynthesis inhibitor) were dissolved in distilled water individually to give a concentration of 1 mg/L of each. Each soil water-holding capacity condition for EI and EF seedlings was assigned to 30 pots, and the leaves of seedlings of 10 pots were sprayed with ABA solution, ten pots were sprayed with fluridone (FLU) solution, and the remaining ten pots were sprayed with distilled water as the control (CK). A total of 12 treatments, and ten replicates per treatment were used in this experiment. Every afternoon of the trial at six o’clock, each pot was weighed, and water was added to maintain the required SWC. After watering, the position of every pot was changed randomly. Foliar spraying of every plant was carried out twice at seven-day intervals with a constant volume of 7 mL/plant of ABA or FLU solution with a manual sprayer at the 6–7 leaf-growth stage. Control plants were sprayed similarly with an equivalent amount of distilled water [49]. All solutions included 0.002% tween-20 and 0.1% ethanol (a minimum amount to dissolve the FLU and ABA) to promote the uptake of ABA and FLU [50]. After 4 weeks, chlorophyll content, photosynthetic indexes, and chlorophyll fluorescence were determined.

### 2.3. Measurement Protocols

#### 2.3.1. Chlorophyll Content

At the end of this trial, chlorophyll content was measured by selecting the top mature leaf of each DHG plant in each treatment using a SPAD-502Plus (Konica Minolta Sensing Inc., Tokyo, Japan), according to the method of Ye et al. [51].

#### 2.3.2. Photosynthetic Indexes

The measurement of photosynthetic indexes was carried out on a single day along with a chlorophyll content measurement. Photosynthesis in the top mature leaf, that is the same leaf of which chlorophyll content was measured, of plants in each treatment was measured by a portable fluorescence system (GFS-3000, WALZ, Effeltrich, Germany) in a chamber between 9:00 and 11:00 a.m. The light saturation point, relative humidity, temperature, air flow rate, and CO_2_ concentration in the chamber were 800 µmol m^−2^ s^−1^, 70%, 25 °C, 750 µmol s^−1^, and 700 mg m^−3^, respectively. Five measurements were performed at five points along each selected leaf on every plant.

#### 2.3.3. Biomass Production and Growth

Growth indices, including the tiller number and plant height, were measured as follows. All of the plants from all of the treatments were harvested and separated into roots and shoots, then washed with distilled water. One plant of each pot of every treatment was oven-dried at 80 °C until a constant weight was reached. The dry weights of roots and shoots of the EF and EI plants were measured.

#### 2.3.4. Statistical Analysis

The statistical data analysis was conducted by the SPSS software, Version 20.0 (SPSS, Inc., Chicago, IL, USA). The effects of ABA, water content, and endophyte were evaluated through multivariate-measures ANOVA; the statistical significance was considered at the 95% confidence level. Means and their standard error are reported. Origin 9.0 (Origin Lab, Northampton, MA, USA) was used to conduct mapping. We also estimated the strength of indirect and direct relationships among the considered variables, including endophyte, ABA, water, stomatal conductance, photosynthetic rate, chlorophyll content, and biomass. To ensure that the linear models were appropriate, the bivariate relationships between variables were first checked. The potentially causal linkages between the different factors and biomass were identified by structural equation modeling (SEM). SEM steps were carried out with AMOS 25.0 (Amos Development Co., Greene, ME, USA).

## 3. Results

### 3.1. Plant Growth Parameters of EI and EF Plants

#### 3.1.1. Plant Height and Tiller Number

The height of DHG plants was significantly (*p* < 0.05) affected by ABA, *Epichloë*, water content × ABA, and ABA × *Epichloë* (Table 1). At the 20% SWC condition, compared with the CK treatment, spraying with ABA increased the plant height by 6.3%, while spraying with FLU significantly (*p* < 0.05) decreased the plant height by 13.4%. EI plants were significantly (*p* < 0.05) higher than EF plants under ABA, FLU, and CK treatments, with a relative difference of 25.0%, 16.4%, and 13.5%, respectively. At the 60% SWC condition, compared with the CK, spraying with ABA significantly (*p* < 0.05) increased the plant height by 12.8%, while spraying with FLU significantly (*p* < 0.05) decreased the plant height by 16.8%. EI plants were also significantly (*p* < 0.05) higher than EF plants under the ABA, FLU, and CK treatments by 33.7%, 14.6%, and 18.0%, respectively. The differences in value of the plant height between EI and EF plants were biggest when sprayed with ABA than with FLU and CK under both the 20% and 60% water conditions (Figure 1).

Water content, ABA, *Epichloë,* and water content×ABA all significantly (*p* < 0.05) affected the tiller number of seedlings (Table 1). At the 20% SWC condition, compared with CK, spraying with ABA significantly (*p* < 0.05) increased the tiller number by 21.5%, while spraying with FLU decreased the tiller number by 10.1%. EI plants were significantly (*p* < 0.05) higher than EF plants under the ABA, FLU, and CK treatments, with a relative difference of 23.3%, 21.9%, and 19.4%, respectively. At the 60% SWC condition, compared with the CK treatment, spraying with ABA significantly (*p* < 0.05) increased the tiller number by 38.9%, while spraying with FLU significantly (*p* < 0.05) decreased the tiller number by 24.4%. EI plants were also significantly (*p* < 0.05) higher than EF plants under the ABA, FLU, and CK treatments, with a relative difference of 4.9%, 0%, and 14.3%, respectively (Figure 1).

#### 3.1.2. Biomass

The shoot dry biomass of DHG was significantly (*p* < 0.05) affected by the water content, ABA, and *Epichloë* (Table 1). At the 20% SWC condition, compared with the CK treatment, spraying with ABA increased the shoot dry biomass by 6.84%, while spraying with FLU significantly (*p* < 0.05) decreased the shoot dry biomass by 14.16%. The dry biomass of EI plants was significantly (*p* < 0.05) higher than EF under the ABA, FLU, and CK treatments, with a relative difference of 24.94%, 22.68%, and 22.81%, respectively. At the 60% SWC condition, compared with the CK treatment, spraying with ABA significantly increased the shoot dry biomass by 17.43%, while spraying with FLU significantly (*p* < 0.05) decreased the shoot dry biomass by 23.00%. EI plants also had significantly (*p* < 0.05) more shoot dry biomass than EF plants under the ABA, FLU, and CK treatments, with a relative difference of 40.97%, 24.23%, and 32.16%, respectively. The differences in value of the shoot biomass between EI and EF plants were biggest when sprayed with ABA, compared with FLU and CK under both the 20% and 60% water conditions (Figure 2).

The root dry biomass of DHG plants was significantly (*p* < 0.05) affected by the water content, ABA, *Epichloë,* and water content × ABA (Table 1). At the 20% SWC condition, compared with the CK treatment, spraying with ABA significantly (*p* < 0.05) increased the root dry biomass by 14.71%, while spraying with FLU significantly (*p* < 0.05) decreased the root dry biomass by 10.89%. EI plants had significantly (*p* < 0.05) lower root dry biomass than EF plants under the ABA, FLU, and CK treatments, with a relative difference of 1.73%, 7.06%, and 10.45%, respectively. At the 60% SWC condition, compared with the CK treatment, spraying with ABA increased the root dry biomass by 4.51%, while spraying with FLU significantly (*p* < 0.05) decreased the root dry biomass by 27.15%. The endophyte also significantly (*p* < 0.05) decreased the root dry biomass under the ABA, FLU, and CK treatments, with a relative decrease of 5.89%, 15.18%, and 14.88%, respectively (Figure 2).

### 3.2. Photosynthetic Indexes of EI and EF Plants

#### 3.2.1. Chlorophyll Content

The chlorophyll content of DHG plants was significantly (*p* < 0.05) affected by the water content, ABA, *Epichloë*, water content × ABA, and ABA × *Epichloë* (Table 2). At the 20% SWC condition, compared with the CK, spraying with ABA increased the chlorophyll content by 16.3%, while spraying with FLU significantly (*p* < 0.05) decreased the chlorophyll content by 23.2%. EI plants had a significantly (*p* < 0.05) higher chlorophyll content than EF under the ABA, FLU, and CK treatments, with a relative difference of 68.0%, 32.5%, and 47.0%, respectively. At the 60% SWC condition, compared with the CK, spraying with ABA increased the chlorophyll content by 9.3%, while spraying with FLU significantly (*p* < 0.05) decreased the chlorophyll content by 39.1%. Endophyte also significantly (*p* < 0.05) increased the chlorophyll content of DHG under the ABA, FLU, and CK treatments, by 54.7%, 34.6%, and 59.1%, respectively. The differences in value of the chlorophyll content between EI and EF plants were biggest when sprayed with ABA compared with FLU and CK under the 20% water condition (Figure 3).

#### 3.2.2. Photosynthetic Rate

The water content, ABA, *Epichloë*, water content × ABA, and water content × ABA × *Epichloë*, all significantly (*p* < 0.05) affected the photosynthetic rate of DHG (Table 2). At the 20% SWC condition, compared with the CK treatment, spraying with ABA increased the photosynthetic rate by 11.2%, while spraying with FLU significantly (*p* < 0.05) decreased the photosynthetic rate by 50.6%. Endophyte increased the photosynthetic rate of DHG under the ABA, FLU, and CK treatments, with a relative difference of 27.2%, 16.4%, and 26.2%, respectively. At the 60% SWC condition, compared with the CK treatment, spraying with ABA decreased the photosynthetic rate by 12.4%, while spraying with FLU also significantly (*p* < 0.05) decreased the photosynthetic rate by 47.6%. EI plants also had significantly (*p* < 0.05) higher photosynthetic rates than EF plants under the ABA, FLU, and CK treatments, with a relative difference of 21.7%, 3.8%, and 46.0%, respectively. The differences in value of the photosynthetic rate between EI and EF plants were biggest when sprayed with ABA compared with FLU and CK under the 20% water condition (Figure 4).

#### 3.2.3. Transpiration Rate

The transpiration rate of DHG plants was significantly (*p* < 0.05) affected by the ABA, water content, *Epichloë*, water content × ABA, and ABA × *Epichloë* (Table 2). At the 20% SWC condition, compared with the CK treatment, spraying with ABA decreased the transpiration rate by 3.4%, while spraying with FLU also decreased the transpiration rate by 17.8%. Endophyte increase the transpiration rate of DHG plants under the ABA, FLU, and CK treatments, with a relative increase of 37.2%, 23.8%, and 54.0% (*p* < 0.05), respectively. At the 60% SWC condition, compared with the CK, spraying with ABA increased the transpiration rate significantly by 11.5%, while spraying with FLU significantly (*p* < 0.05) decreased the transpiration rate by 28.5%. EI plants also had significantly (*p* < 0.05) higher transpiration rates than EF plants under the ABA, FLU, and CK treatments, with a relative difference of 24.6%, 15.5%, and 28.4%, respectively (Figure 4).

#### 3.2.4. Stomatal Conductance

Water content, ABA, *Epichloë,* and ABA × *Epichloë*, significantly (*p* < 0.05) affected the stomatal conductance of DHG plants (Table 2). At the 20% SWC condition, compared with the CK treatment, spraying with ABA significantly (*p* < 0.05) increased the stomatal conductance by 35.7%, while spraying with FLU significantly (*p* < 0.05) decreased the stomatal conductance by 27.7%. Endophyte significantly (*p* < 0.05) increased the stomatal conductance of DHG plants under the ABA, FLU, and CK treatments, with a relative increase of 42.9%, 31.1%, and 46.2%, respectively. At the 60% SWC condition, compared with the CK, spraying with ABA significantly (*p* < 0.05) increased the stomatal conductance by 31.9%, while spraying with FLU significantly (*p* < 0.05) decreased the stomatal conductance by 22.0% (*p* < 0.05). EI plants also had significantly (*p* < 0.05) higher stomatal conductance than EF plants under the ABA, FLU, and CK treatments, with a relative difference of 39.5%, 38.2%, and 50.5%, respectively (Figure 4).

#### 3.2.5. Intercellular Carbon Dioxide Concentration

The intercellular carbon dioxide concentration of DHG was significantly (*p* < 0.05) affected by water content, ABA, *Epichloë*, water content × ABA, ABA × *Epichloë*, water × *Epichloë*, and water content × ABA × *Epichloë* (Table 2). At the 20% SWC condition, compared with the CK treatment, spraying with ABA significantly (*p* < 0.05) increased the intercellular carbon dioxide concentration by 27.5%, and spraying with FLU also significantly (*p* < 0.05) increased the intercellular carbon dioxide concentration by 49.4%. EI plants had significantly (*p* < 0.05) lower intercellular carbon dioxide concentrations than EF plants under the ABA, FLU, and CK treatments, with a relative difference of 13.0%, 9.6%, and 40.9%, respectively. At the 60% SWC condition, compared with the CK, spraying with ABA significantly (*p* < 0.05) increased the intercellular carbon dioxide concentration by 24.7%, spraying with FLU also significantly (*p* < 0.05) increased the intercellular carbon dioxide concentration by 68.0%. Endophyte decreased the intercellular carbon dioxide concentration of DHG plants significantly (*p* < 0.05) under the ABA, FLU, and CK treatments by 10.2%, 9.1%, and 28.9%, respectively (Figure 4).

### 3.3. SEM for the Interactive Effects of Endophyte, Water and ABA on DHG Biomass

SEM analyses confirmed that the interactive effects of endophyte, soil water content and spraying with ABA affected plant biomass via the pathway of photosynthesis indicators. The best SEM adequately fitted the data: χ^2^ = 0.662, *p* = 0.956, RMSEA < 0.001, GFI = 0.997, AGFI = 0.978, RFI = 0.992 and was successful in explaining the variance in plant biomass (R^2^ = 90%). In the final model, the biomass was negatively affected by endophyte (*p* < 0.01) directly, while spraying with ABA (*p* < 0.01) and SWC (*p* < 0.001) positively affected the biomass directly. The final model explained 87%, 80%, and 84% of variation in stomatal conductance, net photosynthetic rate, and chlorophyll content. Additionally, endophyte, spraying with ABA, and SWC also indirectly affected plant biomass positively via the above-mentioned three indicators, with most of the pathways being significant (*p* < 0.05) (Figure 5).

## 4. Discussion

The key findings of the present study are that, compared with the CK, spraying with ABA increased the growth and photosynthetic capacity of both EI and EF DHG plants, and it was a reversed pattern when applying the ABA biosynthesis inhibitor FLU. Critically, the positive effects of endophyte on growth and photosynthesis were the largest when sprayed with ABA, followed by spraying with distilled water, with the lowest positive effect occurring when sprayed with FLU. These effects of ABA and FLU occurred at both the high and low SWC. These findings indicate that ABA might be one of the substances that leads to the enhanced growth of DHG grasses that are hosts to an *Epichloë* endophyte, under both moist and dry soil moisture conditions.

The findings of the study provide support for the hypothesis that ABA enhances the growth of DHG plants. The present results showed that spraying ABA further increased the values of the tiller number, plant height, root dry biomass, and shoot dry biomass of EI DHG and similarly increased the values of EF plants. Similarly, the application of ABA enhanced the values of the chlorophyll content, stomatal conductance, transpiration rate, and net photosynthetic rate of EI plants, with similar changes happening with EF plants. Compared with EF plants, the application of ABA enhanced the differences in indicators of plant height, shoot biomass, chlorophyll content, and the photosynthetic rate of EI plants. Interestingly, the positive effects to the presence of the endophyte were greater at 60% WSC than 20% WSC. Supportive of the role of ABA, the application of the ABA suppressor FLU had the reverse effect of ABA on the nine indicators of growth and photosynthesis in this study.

Drought inhibits the growth of plants, but fungal endophytes could increase host plant biomass [52,53,54]. In the present study, it was found that the effects of the *Epichloë* endophyte on host biomass under drought stress were divided into direct and indirect aspects. The direct effects of *Epichloë* endophyte on host biomass were negative, as the *Epichloë* hyphal biomass is very small compared with the plant biomass, but hyphae remain metabolically active for the life of the tissue that they are in, which would be an ongoing cost of photosynthetic products and the energy of the host [55]. The indirect effects of *Epichloë* endophyte on host biomass were positive by alleviating drought damage to the photosynthetic capacity [27]. Obviously, the indirect effect is greater than the direct effect of *Epichloë* in the present study, as the shoot biomass of EF DHG is significantly lower than that of EI DHG. Differently from shoot biomass, the root dry biomass of EF plants was more than those with endophyte. This is different from several previous studies that reported that the presence of *Epichloë* endophyte could increase root dry matter for improving nutrient and water uptake from the soil to enhance drought tolerance [1]. However, a more recent study indicated that the positive effect of *Epichloë* endophyte on hosts exposed to limited water conditions may not only be by absorbing more water via a more developed root system, but also increasing the use efficiency of a limited water resource [27]. This increase in water use efficiency may be why EI plants do not allocate additional resources to root development.

The gain of biomass depends on respiration, photosynthesis, and the allocation of assimilates [56]. More biomass attributes to more photosynthetic products accumulated in plants to enhance survival through the abiotic stress condition [57]. Previous studies have proved that the *Epichloë* endophyte not only directly improves the photosynthetic indexes of host plants [27,58], but also indirectly increases chlorophyll concentration to improve photosynthesis [58], although the detailed mechanism is still unknown. Chlorophyll is the main site of plant photosynthesis and is easily damaged by drought stress [59]. *Epichloë* endophytes can effectively alleviate the loss of chloroplasts caused by drought stress [60]. Stomatal closure is a critical tool for vascular plant survival, preventing plant dehydration and minimizing transpiration [61]. Previous studies on tall fescue and meadow fescue (*F. pratensis*) found that the *Epichloë* endophyte can promote stomatal closure of host grasses and effectively retain plant water in order to have a strong competitiveness in the process of drought tolerance [62]. However, the endophyte also negatively affected the plant growth rate since it reduces carbon dioxide (CO_2_) uptake, which is closely related to the efficiency of photosynthesis [63]. In the present study, the results showed that EI plants had higher stomatal conductance, benefiting the exchange of CO_2_ and increasing of the photosynthetic efficiency [64]. This may be because DHG plants change their metabolism from defense to growth under drought stress, which allows grasses to utilize available water resources for photosynthesis and carbon assimilation—only in this way were plants able to allocate more resources to shoots to develop inflorescences and complete seed production before the most intense period of drought [65]. Photosynthesis shapes the symbiotic relationships between *Epichloë* endophyte and the host grass, as these biotrophic endophytes require the photosynthates of the host grass for survival. Therefore, the improvement of the photosynthetic rate might be due to an increase in nutrition and energy demand to “feed” the fungal partner under stress conditions [66]. On the other hand, the cross-talk between associated the endophyte and plant might be developed during long term co-evolution. The growth of the *Epichloë* species is fully synchronized with the host grass. The host plants are symptomless with the exception of stromata that can form on reproductive tillers of grasses infected with some *Epichloë* spp. Importantly, the plant hosts for *Epichloë* endophytes are highly competitive and not stunted. Thus, the host plant must enhance photosynthesis, otherwise it will become stunted and unable to compete with other plants. Thus, it is not hard to understand that the *Epichloë* endophyte significantly enhanced host biomass by improving the photosynthetic capacity under drought stress.

An important insight into the probable enhancement of grass hosts to an *Epichloë* endophyte comes from a previous study which reported that nearly all naturally growing tall fescue plants in New Zealand, as well as many plants of populations of perennial ryegrass (*Lolium perenne*), are hosts to an *Epichloë* endophyte, while most also have another seed-borne systemic endophyte [67]. The important thing about these endophytes, now referred to as ‘P-endophytes’, is that their growth is not synchronized with that of the host and as leaves age they contain increasing amounts of hyphae, and these hyphae are full of lipid droplets [68]. This non-synchronized growth and lipid production must surely come at a cost to the plant and yet the dually infected plants are indistinguishable from EI-only plants. Thus, either the ‘P-endophyte’ provides a competitive advantage or else photosynthesis is enhanced to cover the cost. Further, another interesting, related finding is that, in comparison with EF DHG, the infection by *Bulmeria graminis*, an obligate biotrophic fungal pathogen, enhanced the photosynthesis of EI DHG plants [69]. Thus, maybe any biotrophic fungus of plants may enhance photosynthesis.

The plant hormone ABA might function as an interspecies communication signal in the natural ecosystems [70]. Not only the primary and secondary metabolism of the *Epichloë* endophyte, but also the fungal growth and development, were all affected by ABA [71]. Besides, it also plays a pivotal role during plant growth and development [72]. Thus, we speculate that there is a potential fungus-plant cross-talk through ABA to improve the competitive advantage or photosynthesis of host grasses. The present results showed that spraying ABA increased the values of the tiller number, plant height, root dry biomass, and shoot dry biomass of EI DHG and similarly increased the values of EF plants, which is similar with exogenous ABA, promoting the shoots dry matter accumulation of *Ilex paraguariensis* and wheat under drought stress [34,49]. The increase in plant growth parameters might be attributed to the positive effects of ABA on the biochemical and metabolic processes, which is associated with the normal growth and development of the plant [73]; for example, the effect of ABA on shoot growth may be attribute to an antagonistic interaction with ethylene, as ethylene is usually inhibitory to the shoot growth of plants [74]. Spraying with FLU led to the insufficiency of endogenous ABA accumulation, with the level of ABA not being sufficient to prevent ethylene-induced shoot growth inhibition [75]. Besides, it was illustrated that the excess ethylene production was a major cause of shoot growth inhibition in wild-type tomato plants by using ethylene-deficient transgenic plants [76]. Further, the previous study reported that a supplemental ABA treatment of the wild-type plants partly reduced shoot ethylene production and restored shoot growth [77]. The common function of endogenous ABA is the restriction of ethylene production. Plants need more ABA to prevent ethylene-induced growth inhibition under water stress. As a result, ABA accumulation may often function to help maintain the shoot as well as root growth during water stress [78]. However, the differences in values of root dry biomass between EI and EF were lowest with the spraying of ABA, indicating that the growth enhancing effect of ABA improved the root biomass of EI plants more than for EF plants. Moderate ABA accumulation stimulates auxin transport (from shoot to root) by increasing the abundance of an auxin efflux carrier, which can enhance basipetal auxin flux, further activating the membrane H^+^- ATPase and thus pumping more H^+^ towards the cell wall, and accumulating H^+^ in the apoplast, increasing cell wall acidification and consequently leading to primary root cell elongation, according to the “acid growth theory” [79]. In this case, it is not hard to understand why exogenous ABA effectively enhanced the root biomass of DHG.

ABA content is closely connected to chlorophyll synthesis in plants under drought stress [80], mediating a higher pool of pigments from the xanthophyll cycle during water stress [81]. The present results are in agreement with an early study that an appropriate concentration of exogenous ABA can increase chlorophyll content under drought stress, which further improved the photosynthesis; this result is also consistent with wheat (*Triticum aestivum*) and perennial ryegrass [50,60]. As a positive regulator of stomatal closure, ABA is one of the substances that leads to the increased drought tolerance of plants. An early study has shown that exogenous ABA increased stomatal closure in plants, likely increasing moisture retention in water deficit conditions [82]. However, the present results showed that spraying with ABA increased the stomatal conductance of DHG, which was contrary to previous research results [82,83]. The modulation of stomatal apertures by ABA is associated with multiple cascades of cellular biochemical events, and one of them is the production of reactive oxygen species (ROS) [84]. For the closure of stoma in response to ABA, guard cells must generate enough ROS. If the ROS production is blocked, ABA-induced stomatal closure is inhibited [85]. However, the presence of *Epichloë* endophyte is able to decrease the levels of ROS content of host plants in part by maintaining the higher enzyme activity of SOD and APX under drought [86]. This may balance out the effects of ABA on stomatal closure.

Previous research had indicated that an *Epichloë* infection improves the accumulation of the phytohormone ABA to improve the tolerance of abiotic and biotic stresses [35,87,88] and the expression of genes encoding key enzymes involved in signaling and biosynthesis by ABA were also up-regulated in EI plants [45]. However, there have been few studies exploring if and how ABA works in this process. The present study expands our knowledge of the relationship between ABA and *Epichloë* endophyte in response to adequate water availability and water deficit conditions. The results demonstrated that the promotion of biomass accumulation of DHG by the *Epichloë* endophyte is achieved through indirectly effecting chlorophyll content, stomatal conductance, and the photosynthetic rate. More importantly, ABA plays a pivotal role in improving the drought tolerance of DHG, and could be one of the potential substances involved in the moderating effect of *Epichloë* endophyte on DHG through enhancing photosynthesis and growth under water deficit condition, as the differences in values of the plant height, shoot biomass, chlorophyll content, and photosynthetic rate between EI and EF were biggest when sprayed with ABA compared with the CK and FLU treatments. However, more research is needed to identify and fully understand the mechanisms of *Epichloë* endophyte utilization of ABA to enhance tolerance to drought stress.

## Figures and Tables

**Figure 1 jof-08-01140-f001:**
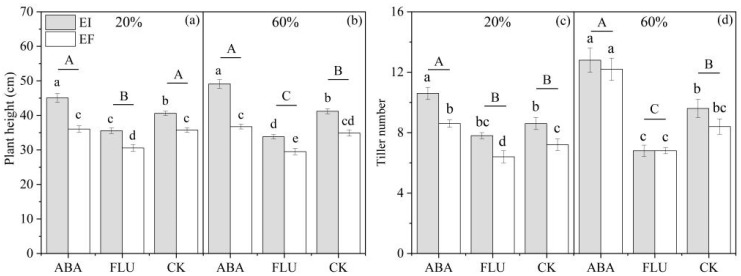
Plant height and tiller number of *A. inebrians* seedlings, endophyte-uninfected (EF) and infected (EI), under different exogenous ABA content and two soil water contents. Columns with non-matching lowercase letters indicate a significant difference between different treatments under the same water content at *p* < 0.05. Columns with non-matching capital letters indicate a significant difference between different ABA treatments under the same water content at *p* < 0.05. The same below. (**a**,**c**): 20% water content, (**b**,**d**): 60% water content.

**Figure 2 jof-08-01140-f002:**
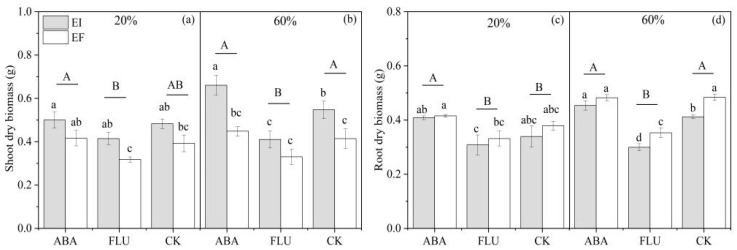
Shoot dry biomass and root dry biomass of two different soil water contents of *A. inebrians* seedlings, endophyte-uninfected (EF) and infected (EI) under different exogenous ABA content (g). (**a**,**c**): 20% water content, (**b**,**d**): 60% water content.

**Figure 3 jof-08-01140-f003:**
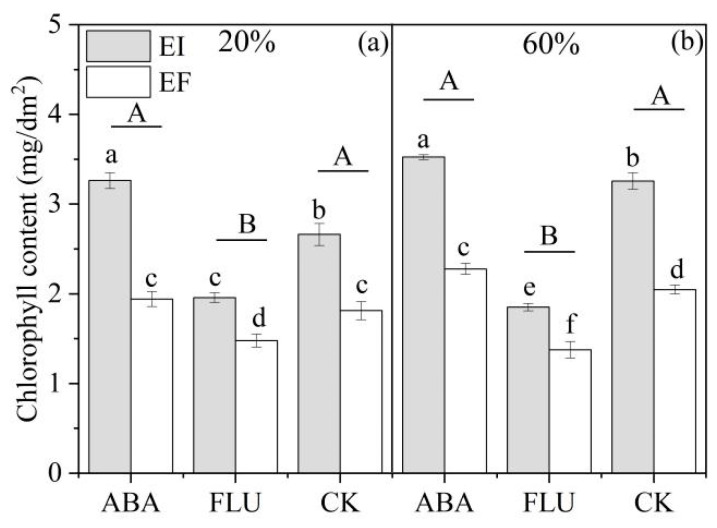
Chlorophyll content of two different soil water contents of *A. inebrians* seedlings, endophyte-uninfected (EF) and infected (EI) under different exogenous ABA content (mg/dm^2^). (**a**): 20% water content, (**b**): 60% water content.

**Figure 4 jof-08-01140-f004:**
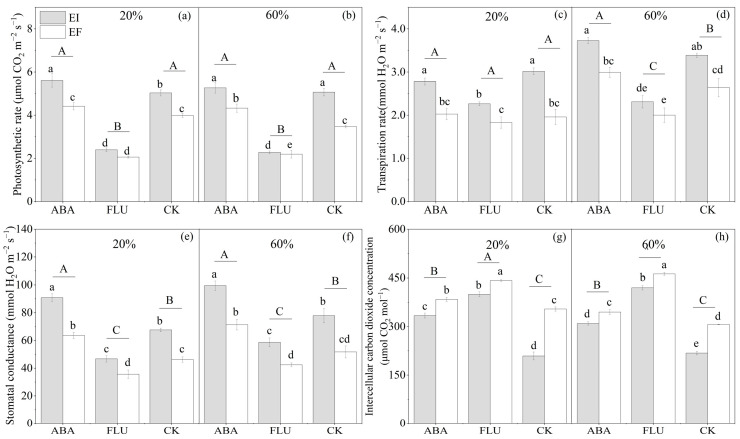
Photosynthetic rate (μmol CO_2_ m^−2^s^−1^), transpiration rate (mmol H_2_O m^−2^s^−1^), stomatal conductance (mmol H_2_O m^−2^s^−1^), and intercellular carbon dioxide concentration (μmol CO_2_ mol^−1^) of endophyte-uninfected (EF) and infected (EI) *A. inebrians* seedlings, at two soil water contents and different exogenous ABA contents. (**a**,**c**,**e**,**g**): 20% soil water content, (**b**,**d**,**f**,**h**): 60% soil water content.

**Figure 5 jof-08-01140-f005:**
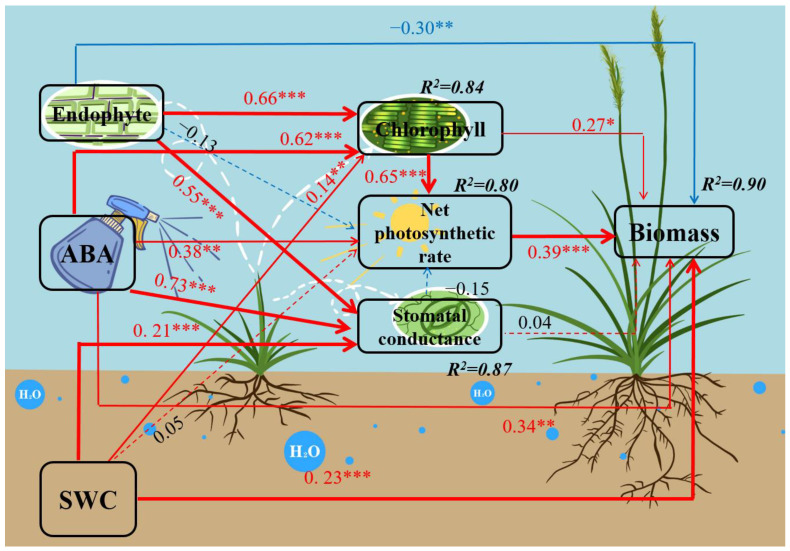
The final model results of structural equation modeling (SEM) analysis for endophyte, abscisic acid, and water factors on biomass of *A. inebrians* seedlings. Dotted arrows, insufficient statistical evidence for path coefficients (*p* > 0.05). Blue arrows indicate evidence for negative relationships, and red arrows indicate evidence for positive relationships. The proportion of variance explained (R^2^) appears alongside each response variable in the model. Width of the arrows shows the strength of the causal relationship, and the numbers adjacent to arrows are standardized path coefficients, which reflect the extent of the relationship, while asterisks indicate statistical significance (*** *p* < 0.001; ** *p* < 0.01; * *p* < 0.05). The final model adequately fitted the data: χ^2^ = 0.662, P = 0.956, RMSEA < 0.001, GFI = 0.997, AGFI = 0.978, RFI = 0.992.

**Table 1 jof-08-01140-t001:** Results of three-way ANOVA for the effects of soil water content (W), ABA treatment (A), and *Epichloë* (E) on plant height, tiller number, shoot dry biomass, and root dry biomass of *A. inebrians* seedlings.

Treatments	df	Plant Height		Tiller Number		Shoot Dry Biomass		Root Dry Biomass	
		F	*p*	F	*p*	F	*p*	F	*p*
W	1	0.322	0.573	20.281	<0.000	5.684	0.021	17.957	<0.000
A	2	114.111	<0.000	76.504	<0.000	16.278	<0.000	33.640	<0.000
E	1	185.884	<0.000	16.133	<0.000	33.324	<0.000	9.857	0.003
W × A	2	4.709	0.014	11.437	<0.000	1.774	0.181	4.063	0.023
W × E	1	1.787	0.188	3.333	0.074	1.630	0.208	1.377	0.246
A × E	2	13.561	<0.000	0.533	0.59	0.742	0.481	0.852	0.433
W × A × E	2	1.302	0.281	0.533	0.59	1.093	0.343	0.021	0.980

**Table 2 jof-08-01140-t002:** Results of three-way ANOVA for the effects of soil water content (W), ABA treatment (A), and *Epichloë* (E) on chlorophyll content, photosynthetic rate, transpiration rate, stomata conductance, and intercellular dioxide concentration of *A. inebrians* seedlings.

Treatments	df	Chlorophyll Content		Photosynthetic Rate		Transpiration Rate		Stomatal Conductance		Intercellular Dioxide Concentration	
		F	*p*	F	*p*	F	*p*	F	*p*	F	*p*
W	1	20.383	<0.001	2.456	0.124	0.016	0.901	1.383	0.245	5.150	0.028
A	2	205.192	<0.001	294.803	<0.001	15.975	<0.001	23.799	<0.001	252.561	<0.001
E	1	425.305	<0.001	82.563	<0.001	73.141	<0.001	59.181	<0.001	225.034	<0.001
W × A	2	12.189	<0.001	0.679	0.512	7.651	<0.001	8.840	<0.001	14.606	<0.001
W × E	1	1.085	0.303	0.005	0.944	4.648	<0.001	3.850	0.056	6.923	0.011
A × E	2	27.756	<0.001	12.402	<0.001	44.930	<0.001	46.957	<0.001	12.100	<0.001
W × A × E	2	2.223	0.119	1.989	0.148	3.324	0.044	2.883	0.066	25.648	<0.001

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
