# Peer review of "Abscisic Acid May Play a Critical Role in the Moderating Effect of Epichloë Endophyte on Achnatherum inebrians under Drought Stress"

_jof, 2022, doi:10.3390/jof8111140_

Round 1
Reviewer 1 Report
General
This manuscript reports interactions among treatments involving abscisic acid, water status, and endophyte status of Achnatherum inebrians (drunken horse grass). The results are interesting and valuable. The manuscript is, however, in need of careful editing to improve the presentation and adhere to standard English style. I have included some examples where the presentation should be improved, but I stopped making detailed suggestions about word choice and sentence structure at approximately line 200. The authors might need to seek some outside help to improve grammar and style.
The Discussion section seems unnecessarily long. The last half of the Discussion reads more like a review article than an original research article. I suggest the authors rewrite the Discussion focusing on points that are directly relevant to the results presented.
Some specific comments
Line 15. “alleviating” rather that “alleviate”
Lines 39-40. Perhaps change “their presence” to “endophyte presence”
Line 42. Change “refers to” to “is”
Line 43. I suggest breaking this into two sentences by changing “Poaceae family, and is” to “Poaceae family. It is”
Line 46. Change “infected” to “infected with”
Line 48. Delete “an”
Line 48. Perhaps change “generally proven” to “proposed”
Lines 49-52. This following passage needs some rewording (the lists at the end are not parallel in construction): “plants by conferring increased growth and competitive ability, including protection from being eaten by livestock [17], and insect pests [18,19], pathogenic fungi resistance [20-22], as well as improved abiotic stress tolerance of host plants, including to low temperature [23], heavy metals [24], salt stress [25,26], and in particular, drought [27].”
I suggest:
“plants by conferring increased growth and competitive ability, including protection from livestock grazing [17], insect pests [18,19], and pathogenic fungi [20-22]. Endophytes are also reported to improve abiotic stress tolerance of host plants, including increased tolerance of low temperature [23], heavy metals [24], salt stress [25,26], and in particular drought [27].”
Lines 68-69. The phrase “main root in order to response drought” needs to be reworded. Perhaps “main root in response to drought”
Lines 73-74. Perhaps change “increase the content of phytohormones in tall fescue (Festuca arundinacea), such as indoleacetic acid” to “increase the content of phytohormones such as indoleacetic acid in tall fescue (Festuca arundinacea)”
Lines 78. Perhaps change “of plants host” to “of plant hosts”
Line 87. Delete “was”. Change semicolon after “water” to a comma.
Line 89. Period needed after “[47]”.
Line 96. Change “the infection statues” to “infection status”
Section 2.1. It appears that EF and EI are not defined.
Section 2.2, first paragraph. The relationship between bags and pots is not clear. A drawing would help.
Table 1. The entries in this table are confusing. The headings indicate that height is in cm and mass is in g, but the table seems to give only F and P values. In addition, it is not clear what is being compared. For example, what is meant by effects of water (W)? A table that summarizes the types of treatments and the results would indeed be valuable, and it would probably be best to put this table in the main body of the paper. As is, this table is not very informative, however.
Section 3.1.1. It appears that “CK” has not been previously defined. Also, although “EI” and “EF” were defined in the Abstract, they should be redefined in the text.
Lines 168-169. In the phrase “relative decrease of 6.3% and 13.4% (P<0.05)” it’s not clear what the 6.3% and 13.4% refer to.
Figure 1 legend. The explanation of uppercase and lowercase letters in the context of statistical significance is not clear.
Lines 201-202. In the following passage it is not clear what “6.84% and 14.16%” refer to:
“while spraying with FLU significantly decreased the shoot dry biomass, with a relative decrease of 6.84% and 14.16% 202 (P<0.05), respectively.”
Lines 207-208. Same point as above for “17.43% and 23.00%”.
Reviewer 2 Report
I recommend the publication of the article because scientific experimentation relating to the study of plant hormones and endophytes that can stimulate resistance to abiotic stresses is of particular interest.
The aim and objectives of the article have been stated and are very interesting. The study of methods to increase plant resistance is an important topic, especially to reduce plant mortality and increase yields in agriculture. The work done is certainly of international interest and the format applied is certainly suitable for a research article. The work is original, of particular interest and can certainly stimulate research on this topic. The length of the article is appropriate for the journal and the graphs and tables are clear and easy to understand. The conclusion summarises the aims of the work and future prospects.
Author Response
Thanks for you pretty sure this study and manuscript. You recommend the publication of the article, which will make my thought and study to become available for the international scientific community to read. It gives me much encouragement and confidence.

Round 2
Reviewer 1 Report
Thank you for your careful attention to the revision.